# The Heterogeneous Habitat of Taiga Forests Changes the Soil Microbial Functional Diversity

**DOI:** 10.3390/microorganisms12050959

**Published:** 2024-05-10

**Authors:** Tian Zhou, Song Wu, Mingliang Gao, Libin Yang

**Affiliations:** 1Key Laboratory of Biodiversity, Institute of Natural Resources and Ecology, Heilongjiang Academy of Sciences, Harbin 150040, China; 15894479670@163.com; 2Science and Technology Innovation Center, Institute of Scientific and Technical Information of Heilongjiang Province, Harbin 150028, China; wusong0927@126.com; 3Heilongjiang Huzhong National Nature Reserve, Huzhong 165038, China; zrbhjzhk@163.com

**Keywords:** Da Xing’anling Mountains, *Larix gmelinii*, soil microorganism, functional diversity, important value

## Abstract

The soil contains abundant and diverse microorganisms, which interrelate closely with the aboveground vegetation and impact the structure and function of the forest ecosystem. To explore the effect of vegetation diversity on soil microbial functional diversity in taiga forests, we selected significantly different important values of *Larix gmelinii* as experimental grouping treatments based on plant investigation from fixed plots in Da Xing’anling Mountains. Following that, we collected soil samples and applied the Biolog-ECO microplate method to investigate differences in carbon source utilization, features of functional diversity in soil microorganisms, and factors influencing them in taiga forests. The AWCD decreased as the important value of *Larix gmelinii* grew, and soil microorganisms preferred carboxylic acids, amino acids, and carbohydrates over polymers, phenolic acids, and amines. The Shannon and McIntosh indexes decreased significantly with the increase of the important value of *Larix gmelinii* (*p* < 0.05) and were positively correlated with soil SOC, MBC, C/N, and pH, but negatively with TN, AP, and AN. Redundancy analysis revealed significant effects on soil microbial functional diversity from soil C/N, SOC, AP, MBC, TN, pH, AN, and WC. To sum up, heterogeneous habitats of taiga forests with different important values altered soil microbial functional diversity.

## 1. Introduction

The forest is the largest terrestrial ecosystem on earth, containing a wide variety of species and an intricate ecological structure. Due to frequent human activity and catastrophic weather events caused by global climate change, forest areas and species diversity are gradually declining [1]. Soil microorganisms, the most active part of the biota in forest soil, are essential to sustaining soil fertility, enhancing the efficiency of nutrient transformation, and advancing the substance cycle of forest ecosystems [2,3,4]. Recently, ecologists have focused on the interaction mechanism between plants and microorganisms, especially using the interaction between plants and soil microorganisms to explain the maintenance mechanism of forest species diversity [5]. Relevant studies have pointed out that aboveground plants and soil microorganisms serve as vital bridges between aboveground and underground ecosystems and that the interaction between the two directly regulates species composition and community structures in forests [6,7]. Therefore, understanding the connection between plant communities and soil microbial diversity is necessary for the restoration and sustainable development of forest ecosystems.

Soil microbes, being the main decomposers in soil, can have an indirect effect on plant community growth and succession by regulating soil nutrient availability [8]. Meanwhile, related research has shown that plants can affect the diversity and composition of soil microbial communities by releasing root exudates and varying the number and species of litter [9,10]. The variations in plant diversity will drive changes in soil microbial functional diversity, which is a crucial indicator to assess soil microbial community traits and roles [11]. It has been demonstrated that the functional diversity of culturable bacterial communities is related linearly to plant species richness [12]. Loranger-Merciris et al. [13] found that culturable soil microbial activity, the degree of substrate utilization, and functional diversity were significantly correlated with plant diversity in grassland ecosystems. In contrast, some evidence has argued that above-ground vegetation diversity does not correlate significantly with soil microbial diversity [14,15]. Although the relationship between vegetation diversity and soil microbial functional diversity has been studied, the influencing factors are complex, and the change rules are still unclear.

Daxinganling is the largest distribution area of taiga forests in China, with a monoculture vegetation composition whose tree layer is primarily made up of *Larix gmelinii*, *Betula platyphylla*, and *Populus davidiana*, among which *Larix gmelinii* is the established species [16]. Soil microbial diversity under different forest types, environmental disturbances, and forest succession in the Greater Khingan Mountains has been studied in recent years [17,18,19]. Unfortunately, there are few studies on the effects of vegetation diversity on soil microbial functional diversity. Additionally, important value is a crucial indicator for assessing the ecological adaptation and community status of plant species and has been frequently used in research on plant diversity [20,21]. Therefore, this study investigated the differences in carbon source utilization capacity and changes in soil microbial functional diversity in taiga forests with different important values, aiming at clarifying the effects of vegetation diversity in taiga forests on soil microbial functional diversity and its influencing factors and providing a reference basis for the mechanism of maintaining species diversity in taiga forests.

## 2. Materials and Methods

### 2.1. Overview of the Study Area

This study area is located in Huzhong National Nature Reserve, which is the northernmost part of Daxing’anling region in Heilongjiang Province (51°49′01′′ N~51°49′1′′ N, 122°59′33′′ E~123°00′03′′ E; Figure 1). The site follows a cold-temperate continental monsoon climate, with a large diurnal temperature change and an average annual temperature of −5.3 °C. The winter season is long, with a frost-free period of around 80 days, and the summer season is shorter, with a duration of no more than 30 days [22]. The topography is high in the east and low in the west, with an elevation of 847–974 m. The region has a uniform vegetation composition, mainly including *Larix gmelinii*, *Pinussylvestris* var. *mongolica*, *Betula platyphylla*, *Populus davidiana*, etc., which is a typical cold-temperate coniferous forest in China [23].

### 2.2. Experimental Design and Soil Sample Collection

This study was carried out on 25 hm^2^ permanent large-scale fixed monitoring sample plots in Huzhong National Nature Reserve, where 20 m × 20 m sample plots were set up. All trees in the sample plots were measured for quantity, height, and diameter at the breast height, and the important value of *Larix gmelinii* was calculated. The following is the computing formula [24]:(1)IV=(RD+RP+RH) / 3
where RD stands for relative density, RP for relative dominance, RH for relative height, and IV for the important value of *Larix gmelinii*.

The quadrats exhibiting significant distinctions in the important values of *Larix gmelinii* were chosen as experimental groups, whereas the three quadrats with no significant distinctions were assigned to the same experimental group based on computation results. Thus, a total of 15 sample plots were selected in this study, and five experimental groups were set up. A previous paper has described the important values of *Larix gmelinii* in each group [25]; see Appendix A for details.

Soil samples were taken from the 0–20 cm soil layer in each sample plot by the five-point sampling method. The soil samples were stripped of plant litter and stones, mixed, and then transported back to the laboratory in an ice box. Two portions of the soil samples were taken: one for assessing the soil’s physical and chemical properties, and the other for evaluating the soil’s microbial functional diversity.

### 2.3. Analyzing Soil Chemical and Physical Properties

Soil total nitrogen (TN) was measured by Kjeldahl method; soil available potassium (AK) was determined by flame photometric technique; soil available phosphorus (AP) was examined by molybdenum antimony colorimetric method; soil available nitrogen content (AN) was obtained by alkaline hydrolysis diffusion method; the quantity of soil organic carbon (SOC) was conducted using an elemental analyzer (Multi N/C 2100S, Analytik Jena AG, Jena, Germany); soil microbial biomass carbon (MBC) was evaluated by chloroform fumigation extraction method; soil pH was estimated by pH meter (PB-10, Sartorius, Göttingen, Germany); soil water content (WC) was calculated using the drying method [26,27,28].

### 2.4. Features of Soil Microbial Carbon Source Utilization

The microbial metabolic activities were assessed using Biolog-ECO plates (Biolog Inc., Hayward, CA, USA). Initially, the soil samples were activated at 25 °C for 24 h. After that, a quantity of 10 g of the activated soil samples was weighed, 90 mL of 0.85% sterile NaCl solution was combined, and the mixture was shaken at room temperature for 30 min. The supernatant was diluted to obtain a 10–3 soil suspension and then inoculated into the Biolog-ECO plates. Each group of experiments was repeated three times and incubated continuously at 25 °C for 336 h. During incubation, the absorbance value at a wavelength of 590 nm was measured by a microplate reader (Thermo Fisher Scientific, Waltham, MA, USA) every 24 h [29].

### 2.5. Data Processing and Analysis

The AWCD is an essential indicator for describing the capacity of soil microbes to use carbon sources [30]. The following is the calculation formula:(2)$$\mathrm{AWCD}=\sum_{i=1}^{31}(C_{i}-R)\;/\;31$$
where C_i_ is the absorbance value of each carbon source well at 590 nm; R is the absorbance value of the control well; when C_i_ − R < 0, it is recorded as 0.

The soil microbial functional diversity index, which includes the Shannon–Wiener, Simpson, and McIntosh indices, was computed using the absorbance value of 120 h [30]. The calculation formula is as follows:(3)$$\mathrm{Shannon}-\mathrm{Wiener}\;\mathrm{index}:\;H\;=-\sum_{i=1}^{31}P_{i}\ln P_{i}$$
(4)$$\mathrm{Simpson}\;\mathrm{index}:\;D=1-\sum_{i=1}^{31}P_{i}^{2}$$
(5)$$\mathrm{McIntosh}\;\mathrm{index}:\;U=\sqrt{\sum_{i=1}^{31}n_{i}^{2}}$$
where P_i_ is the ratio of the relative absorbance value of the i-th well to the sum of the relative absorbance values of all wells, and n_i_ is the relative absorbance value of the i-th well.

The differences in soil physical and chemical properties of different important values of taiga forests were compared using SPSS (25.0) by one-way analysis of variance. The soil microbial functional diversity index was summarized and calculated using Excel (2003), and a line graph of soil microbial carbon metabolism activity was plotted. The Heatmap map of the difference in the utilization of carbon sources by soil microorganisms and the correlation analysis of carbon utilization capacity and functional diversity index of soil microorganisms with soil physicochemical factors, respectively, were plotted by R (3.3.1) and python (2.7). The redundancy analysis of soil microbial functional diversity with soil physicochemical factors was performed using the ‘vegan’ package of R (3.3.1) software.

## 3. Results

### 3.1. Differences in Soil Physical and Chemical Properties

The differences in soil physicochemical properties of taiga forests with different important values have been described in the previous article [25], and the relevant data are shown in Appendix A. Here is a brief overview: soil TN, AK, AP, AN, SOC, MBC, pH, WC, and C/N contents were significantly different in taiga forests (*p* < 0.05). Specifically, AP content increased significantly as the important values of *Larix gmelinii* rose, whereas pH, SOC, MBC, and C/N declined significantly (*p* < 0.05).

### 3.2. Differences in the Utilization of Different Carbon Sources by Soil Microorganisms

The AWCD of soil microorganisms in taiga forests rose with prolonged incubation time, as shown in Figure 2. Within the first 24 h of incubation, there was no significant change in AWCD, but a sharp increase was observed after 24 h, with the fastest growth rate occurring between 48 and 120 h. After 120 h, it tended to stabilize, indicating that the utilization of carbon sources by culturable soil microorganisms was primarily concentrated during this stage. Furthermore, soil microbial carbon sources utilization was significantly different at 120 h, which was arranged as L1 > L2 > L4 > L3 > L5, suggesting that as the important value of larch increased, the ability of soil microbial communities to utilize carbon sources decreased.

As shown in Figure 3, soil microorganisms in taiga forests exhibit significant differences in the utilization of carbon sources (*p* < 0.05). Among them, the utilization abilities of D-glucosaminic acid, γ-Hydroxybutyric Acid, D-Malic Acid, L-Asparagine, L-Phenylalanine, N-Acetyl-D-Glucosamine, Tween 40, and Putrescine in L1, L2, and L4 were significantly higher than L3 and L5. The utilization ability of D-galactonic acid γ-lactone in L3 was found to be significantly higher than that of other experimental groups. Conversely, the utilization ability of L-Serine, L-Threonine, Glycyl-L-Glutamic-Acid, and D-Mannitol in L5 was notably lower. Overall, soil microorganisms in each taiga forest exhibited a high utilization of carboxylic acids, amino acids, and carbohydrates, while their utilization of polymers, phenolic acids, and amines was relatively low.

### 3.3. Analysis of Soil Microbial Functional Diversity

To further clarify the changes in soil microbial functional diversity in taiga forest with different important values, an Alpha diversity analysis was performed on the AWCD of cultivated microorganisms for 120 h, as detailed in Table 1. The AWCD, Shannon index, and McIntosh index were significantly different among the groups (*p* < 0.05) and tended to decrease with the important value of *Larix gmelinii* increasing. The Simpson index of L1, L2, and L4 was significantly higher than that of L3 and L5 (*p* < 0.05), and there was no significant change rule.

### 3.4. Correlation between Soil Microbial Functional Diversity and Soil Physical and Chemical Properties

The results of the correlation analysis between soil microbial utilization of different carbon sources and soil physical and chemical factors are shown in Figure 4. D-galacturonic acid, L-Arginine, N-Acetyl-D-Glucosamine were significantly positively correlated with SOC (*p* < 0.001), MBC (*p* < 0.001), pH (*p* < 0.01), and C/N (*p* < 0.001), and significantly negatively correlated with TN (*p* < 0.001), AK (*p* < 0.05), AP (*p* < 0.001), AN (*p* < 0.001), and WC (*p* < 0.05); 2-HydroxyBenzoic Acid, α-Ketobutyric Acid, L-Threonine, Phenylethylamine, and D-Cellobiose were significantly positively correlated with SOC, MBC, pH, and C/N (*p* < 0.01), and significantly negatively correlated with TN (*p* < 0.05), AP (*p* < 0.01), AN (*p* < 0.05), WC (*p* < 0.05); D-glucosaminic acid was significantly negatively correlated with AP (*p* < 0.05); L-Asparagine, L-Phenylalanine, and Tween40 were positively correlated with MBC (*p* < 0.05) and significantly negatively correlated with AP (*p* < 0.05); Itaconic Acid and Glycyl-L-Glutamic Acid was significantly positively correlated with AK (*p* < 0.01); D-Mannitol was significantly positively correlated with MBC (*p* < 0.01)and pH (*p* < 0.05), and significantly negatively correlated with TN (*p* < 0.05), AP (*p* < 0.01), and AN (*p* < 0.05); Putrescine was significantly positively correlated with SOC, MBC, and C/N (*p* < 0.01), and significantly negatively correlated with AP (*p* < 0.01) and WC (*p* < 0.05).

The correlation between the soil microbial functional diversity index and soil physical and chemical factors is shown in Table 2. AWCD and McIntosh index were significantly positively correlated with SOC (*p* < 0.01), MBC (*p* < 0.001), pH (*p* < 0.01), and C/N (*p* < 0.01), and significantly negatively correlated with TN (*p* < 0.01), AP (*p* < 0.001), AN (*p* < 0.01), and WC (*p* < 0.05). Shannon and Simpson indices were significantly positively correlated with SOC (*p* < 0.05), MBC (*p* < 0.01), pH (*p* < 0.01), and C/N (*p* < 0.05), and significantly negatively correlated with TN (*p* < 0.05), AP (*p* < 0.01), and AN (*p* < 0.01).

Redundancy analysis was performed to explore the effects of soil physical and chemical factors on soil microbial functional diversity, and the results are displayed in Figure 5. The RDA1 and RDA2 explained 42.46% and 14.30% of the difference in soil microbial functional diversity, respectively, and the cumulative interpretation rate was 56.76%. The distribution of L2 and L4 in the second quadrant, L1 in the third quadrant, and L3 and L5 in the fourth quadrant showed that the soil microbial carbon source utilization abilities of L2 and L4, L3 and L5, were more similar but differed from other groups. Moreover, RDA1 and RDA2 were positively connected with AK, AN, AP, TN, and WC while negatively correlated with pH, C/N, SOC, and MBC.

It can be seen from Table 3 that all soil physical and chemical factors have an impact on soil microbial functional diversity. With the exception of AK, the following factors exhibited significant effects on soil microbial functional diversity: C/N, SOC, AP, MBC, TN, pH, AN, and WC.

## 4. Discussion

### 4.1. Effects of Heterogeneous Habitats on the Ability of Soil Microbial Carbon Source Utilization in Taiga Forests

Habitat heterogeneity refers to the unequal distribution of resources or other biotic and abiotic environmental factors in its natural environment [31]. The above-ground plant community changes can promote the formation of heterogeneous habitats by affecting root exudates, the types and quantities of litter, and the redistribution of soil moisture and nutrients [32]. Therefore, taiga forests with different important values had significant habitat heterogeneity in this study. The ability of soil microorganisms to utilize carbon sources and metabolic activity is commonly reflected by AWCD [33]. As the culture time was extended, the AWCD of soil microorganisms demonstrated periods of adaptability, fast development, and stability, and it dropped as the important value of *Larix gmelinii* increased. The reason for this result may be that the study area lies in the cold temperate monsoon climate zone, where the growth season of taiga forest is short, the hydrothermal conditions are poor, and the carbon source and nutrient elements supplied to soil microorganisms are comparatively few, thus inhibiting the carbon metabolic activity of soil microorganisms [34]. Besides that, most of the hard-to-decompose coniferous litters in the heterogeneous habitats of taiga forests are not conducive to the accumulation of carbon sources and the acquisition of effective nutrients by the microbial community, leading to a reduction in the AWCD [35].

The differences in soil microbial ability to utilize different carbon sources reflect changes in the functional diversity of soil microbial communities [36]. The results of this study indicated that carboxylic acids, amino acids, and carbohydrates were the main carbon sources of soil microorganisms in taiga forests, which was consistent with the results of Duan et al. [37]. This may be because amino acids and carbohydrates from carbon sources are easily metabolized high-energy substrates that soil microorganisms can exploit with ease [38]. Meanwhile, root secretions provide effective carbon sources for soil microorganisms, and carboxylic acids are the main components secreted by plant roots, so they are widely utilized in soil microorganisms [39]. Moreover, this study found that the utilization of distinct carbon sources by soil microorganisms in taiga forests was significantly different. Chen et al. suggested that variation in soil physicochemical properties is a key factor affecting the ability of soil microorganisms to utilize carbon sources [40], which was also confirmed in this study. The results of correlation analysis revealed that soil microbial utilization ability for different carbon sources was significantly positively correlated with SOC, MBC, pH, and C/N and negatively correlated with TN, AP, and AN. Among them, soil carbon, nitrogen, and phosphorus are the main sources of nutrients required by microorganisms, whose changes in content could affect soil microbial growth and reproduction, ultimately influencing its carbon source utilization intensity [41]. Meanwhile, soil TN and AN can also influence the supply of nutrients to the soil by altering plant root dynamics, changing the pattern of microbial carbon source utilization, and thus significantly affecting the intensity of soil microbial carbon source utilization [42].

### 4.2. Effects of Heterogeneous Habitats on Soil Microbial Functional Diversity in Taiga Forests

The soil microbial functional diversity index is a key indicator for evaluating the status and performance of microbial communities and can reflect changes in the characteristics of microbial communities across various habitats [43]. As a result of this study, AWCD, Shannon index, and McIntosh index decreased significantly as the important value of *Larix gmelinii* increased and were significantly positively correlated with SOC, MBC, pH, and C/N and negatively with TN, AP, and AN. As a consequence, soil nutrients, including SOC, MBC, and C/N, may be significantly reduced in the heterogeneous habitat of taiga forests, limiting the growth and reproduction of soil microorganisms, which in turn reduces their functional diversity [44]. Furthermore, some studies have claimed that a rise in soil AP content may enhance soil microbial functional diversity [45], which was inconsistent with this study. Although soil AP content has significantly increased in the heterogeneous habitat of taiga forests, its content may not reach the level of promoting the growth and metabolism of soil microorganisms, which is still one of the factors restricting the functional diversity of soil microorganisms [46].

As the specific environment in which organisms live, habitat heterogeneity plays an important role in sustaining the diversity of biological species [47]. The results of the RDA analysis found that soil C/N, SOC, AP, MBC, TN, pH, AN, and WC were the main factors significantly shaping soil microbial functional diversity in this study. Relevant studies have pointed out that SOC has a significant effect on soil microbial functional diversity [41], which is consistent with this study. Soil organic carbon is the main source of energy required for soil microorganisms to live, and its content changes directly affect the carbon source utilization efficiency of soil microbial communities [48]. Therefore, the notable drop in SOC in this study significantly affects soil microbial functional diversity. Meanwhile, Soil TN, AP, and AN are the basis for the maintenance of various soil functions and material conversions, and changes in their contents have a significant impact on soil microbial functional diversity [49]. In addition, soil moisture content and pH are important factors affecting soil microbial metabolic activity and diversity. Soil moisture influences the uptake of oxygen and water that soil microorganisms need by controlling soil permeability and humidity, which in turn regulates the functional diversity of soil microorganisms [50]. Weng et al. verified that soil pH is an important driving factor affecting soil microbial carbon uptake, which is consistent with the results of this study [51]. Different soil microbial communities may tolerate varying levels of soil acidity and alkalinity [52], thus the significant decrease in soil pH in the heterogeneous habitat of taiga forests affected soil microbial functional diversity.

## 5. Conclusions

(1)The capacity of soil microbes to utilize carbon sources in taiga forests with different important values was significantly different, with higher utilization of carboxylic acids, amino acids, and carbohydrates than polymers, phenolic acids, and amines.(2)The soil microbial functional diversity decreased significantly with the increase in the important value of *Larix gmelinii*, and soil TN, AP, AN, SOC, MBC, pH, C/N, and WC were the main influencing factors.

## Figures and Tables

**Figure 1 microorganisms-12-00959-f001:**
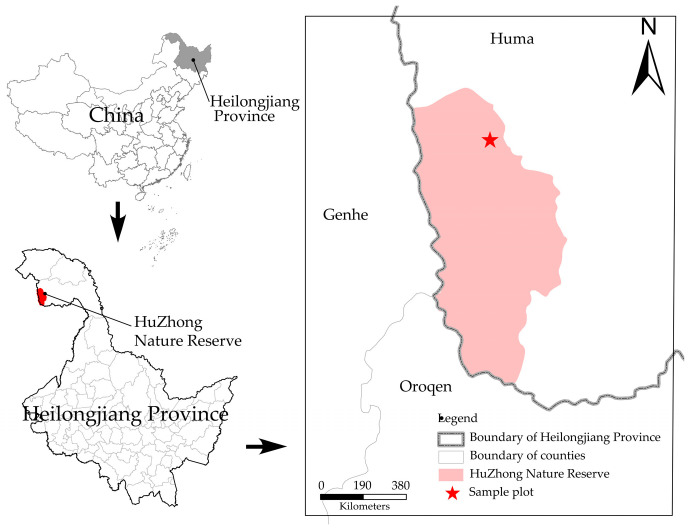
Distribution of research sites in the Daxinganling.

**Figure 2 microorganisms-12-00959-f002:**
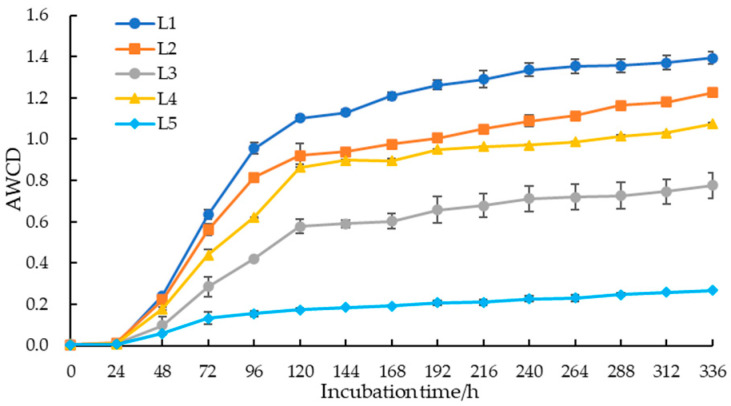
The AWCD of soil microbial community in taiga forests with different important values.

**Figure 3 microorganisms-12-00959-f003:**
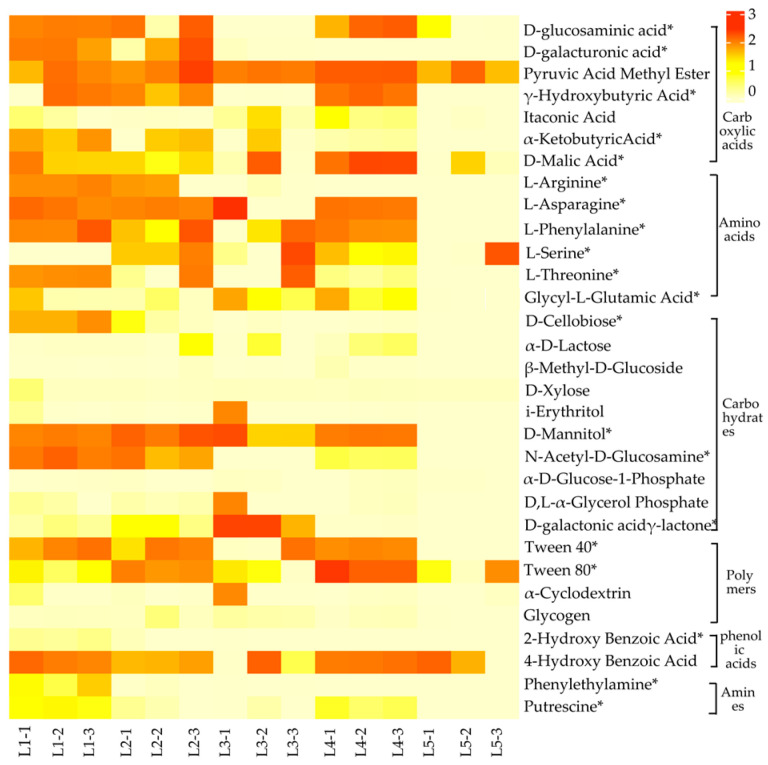
Heatmap of the capacity of soil microorganisms to use carbon sources in taiga forests with different important values. The color gradient represents the soil microbial carbon source utilization rate in taiga forests. The asterisks indicate significant differences in soil microbial ability to utilize carbon sources in taiga forests based on one-way ANOVA at a significance level of 0.05.

**Figure 4 microorganisms-12-00959-f004:**
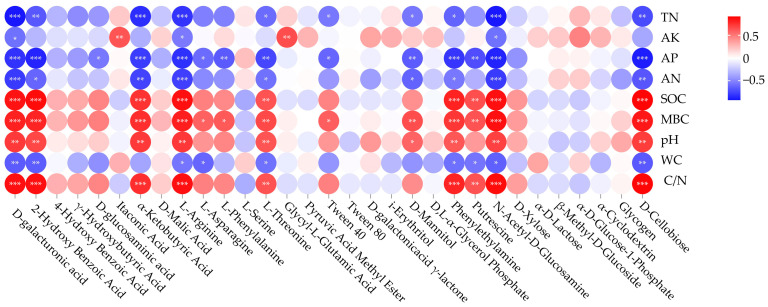
Correlation heatmap of soil microbial utilization of different carbon sources and soil physical and chemical factors. Color shows the intensity of correlation, and red stands for positive, while blue reflects negative correlations. Statistical significance is indicated as *, **, and *** at the level of 0.05, 0.01, and 0.001, respectively.

**Figure 5 microorganisms-12-00959-f005:**
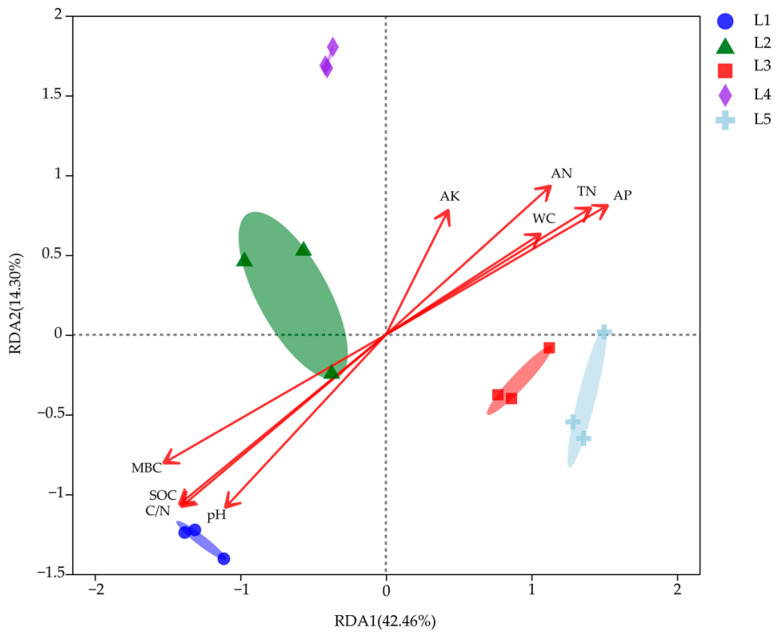
Redundancy analysis of soil microbial functional diversity and soil physical and chemical factors.

**Table 1 microorganisms-12-00959-t001:** The soil microbial functional diversity index in taiga forests. All data are displayed as mean ± standard error (n = 3), and different letters in the same column denote significant difference in taiga forests with different important (*p* < 0.05).

Experimental Group	AWCD	Shannon Index	Simpson Index	McIntosh Index
L1	1.10 ± 0.02 ^a^	3.01 ± 0.03 ^a^	0.95 ± 0.00 ^a^	7.85 ± 0.05 ^a^
L2	0.92 ± 0.06 ^b^	2.87 ± 0.01 ^b^	0.94 ± 0.00 ^a^	7.11 ± 0.41 ^b^
L3	0.58 ± 0.04 ^c^	2.38 ± 0.07 ^c^	0.90 ± 0.01 ^b^	5.75 ± 0.23 ^c^
L4	0.86 ± 0.01 ^b^	2.78 ± 0.01 ^b^	0.93 ± 0.00 ^a^	7.04 ± 0.07 ^b^
L5	0.19 ± 0.00 ^d^	1.30 ± 0.06 ^d^	0.70 ± 0.01 ^c^	3.21 ± 0.10 ^d^

**Table 2 microorganisms-12-00959-t002:** Correlation between soil microbial functional diversity indices and soil physical and chemical factors. Statistical significance is indicated as *, **, and *** at the level of 0.05, 0.01, and 0.001, respectively.

	TN	AK	AP	AN	SOC	MBC	pH	WC	C/N
AWCD	−0.751 **	−0.058	−0.858 ***	−0.690 **	0.732 **	0.865 ***	0.684 **	−0.579 *	0.738 **
Shannon index	−0.595 **	0.196	−0.729 ***	−0.653 **	0.521 *	0.748 ***	0.654 **	−0.414	0.529 *
Simpson index	−0.668 *	0.083	−0.789 **	−0.678 **	0.615 *	0.806 **	0.670 **	−0.463	0.622 *
McIntosh index	−0.692 **	0.054	−0.812 ***	−0.662 **	0.650 **	0.818 ***	0.657 **	−0.550 *	0.658 **

**Table 3 microorganisms-12-00959-t003:** Significance tests between soil microbial functional diversity and soil physical and chemical factors.

Soil Factor	R^2^	*p*-Value
TN	0.806	0.001
AK	0.235	0.196
AP	0.920	0.001
AN	0.658	0.006
SOC	0.966	0.001
MBC	0.918	0.001
pH	0.730	0.003
WC	0.465	0.024
C/N	0.968	0.001

## Data Availability

Data available upon request.

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
