# Peer review of "The Heterogeneous Habitat of Taiga Forests Changes the Soil Microbial Functional Diversity"

_microorganisms, 2024, doi:10.3390/microorganisms12050959_

Round 1

Reviewer 1 Report

Comments and Suggestions for Authors

I found the study topic very interesting and manuscript is generally well prepared. The studied region is very interesting as relatively not so touched by human activity.

There are, however, some minor issues which have to be adressed:

L56, 63 I suggest not to concentrate on scholars, but rather on scientific issue in these sentences, for example would be better to write ‘soil microbial diversity was studied…’

L64-66 I do not agree, there is a plenty of such studies (however, Biolog plates are less popular last years due to development of molecular techniques)

L83 800 days looks strange – it is denote single cold season duration?

L86 what is ‘brown coniferous forest soil’? please use a common classification according to World Reference Base for Soil Resources (WRB)

L86 use uniform instead single

L93 precise how plots were delineated (random or systematic way)

L116-117 soil microbial biomass do not fit to this subchapter about soil physicam and chemical properties. Please add a reference for this method

 L192 there is not so obvious effect as showed in table 1 – I suggest to write that awcd tended to decrease with Larix dominance

L200 Why just 15 substrates were shown of Fig4? Eco plates have 31 substrates. If only some were presented, please specify it in the text

L245 and Table 3 MCB is not soil physical and chemical factor, please remove it form this analysis

L263 cold temperate zone or monsoone zone?

L261 I wonder if altidute above sea level should be included into analysis. Please check if the average altitude differ between Larix groups. The gradient is not big (ca 100m) but these may affect vegetation (i.e. timberline). Moreover, altitude itself can also affect soil microorganisms (please see:  https://doi.org/10.1016/j.pedobi.2015.04.005)

L265 Larix is not a typical coniferous tree as its needles are relatively soft

L270-272 such observation is common for each soil, as these substrates guilds are the most numerous on Biolog Eco plates; please reformulate it

L327 different between what?

Comments on the Quality of English Language

as above

Author Response

Thank you very much for taking the time to review our manuscript. All of these comments are valuable and helpful in revising and improving our paper. We have studied comments carefully and have made corrections to them. Please see the attachment.

Reviewer 2 Report

Comments and Suggestions for Authors

Additional comment:

Title:“The heterogeneous habitat of Taiga forests changes the soil microbial functional diversity”

-       The title corresponds to the content of the paper. 

-        

-       In this study,  the main question is evaluated the differences in carbon source utilization capacity of soil microorganisms and functional diversity as well as influence of the soil physical and chemical properties on carbon utilization, in the taiga forests to establish how taiga forests maintain species diversity. 

-       This study represents a significant contribution to determining composition of microbiala communities and biodiversity of microorganisms and specificity of utilization of distinct carbon sources by soil microorganisms in taiga forests in region with a single vegetation composition, mainly including Larix gmelinii, Pinussylvestris var. mongolica, Betula platyphylla, Populus davidiana, etc., which is a typical cold-temperate coniferous forest in Daxing'anling in China.

-       The obtained results of research represents and expands knowledge about changes of soil microorganisms association due to interaction of soil physical and chemical properties and root of aboveground plants Larix gmelinii and microbial functional difersity.

-       In this paper, the design of the experiment and the applied methods are compatible and the research is rounded. for future new research, focused on the interaction mechanism between plants and microorganisms, especially using the interaction between plants and soil microorganisms to explain the maintenance mechanism of forest species diversity

-       The aim of research  is  not clearly and fully pointed  in abstract. The aim should be clearly pointed in particular  paragraph  out on the end of chapter of Introduction.

-       Key words are appropriate.

-        

-       Sugestion:

-       From lines 68 to 76, it is stated what was the subject of the research, but it is not decisive. The research methods, as well as the possible benefit of the research, are mentioned. This is not allowed in the presentation of the research objective. Methods are reported in the Material and Methods chapter.

-       The benefit of the research can be stated in the discussion or possibly in the conclusion

-       In a scientific paper, the last paragraph of the introduction must clearly state the objective of the research and must be clearly presented

-        

-       The conclusions are clear and based on research results

-       Quoted references are appropriate.

-       Tables, figures, pictures are clear.

-        

-       Manuscript is acceptable after minor corrections!

Author Response

Thank you very much for taking the time to review this manuscript. All of these comments are valuable and helpful in revising and improving our paper. We have studied comments carefully and have made corrections to them. Please see the attachment
